# Thermal Hall effects due to topological spin fluctuations in YMnO$_3$

Ha-Leem Kim [1,2,9], Takuma Saito[3,9], Heejun Yang [1,2,9], Hiroaki Ishizuka [4], Matthew John Coak [1,2,5], Jun Han Lee [6], Hasung Sim[1,2], Yoon Seok Oh [6], Naoto Nagaosa[7] & Je-Geun Park [1,2,8]

The thermal Hall effect in magnetic insulators has been considered a powerful method for examining the topological nature of charge-neutral quasiparticles such as magnons. Yet, unlike the kagome system, the triangular lattice has received less attention for studying the thermal Hall effect because the scalar spin chirality cancels out between adjacent triangles. However, such cancellation cannot be perfect if the triangular lattice is distorted. Here, we report that the trimerized triangular lattice of multiferroic hexagonal manganite YMnO$_3$ produces a highly unusual thermal Hall effect under an applied magnetic field. Our theoretical calculations demonstrate that the thermal Hall conductivity is related to the splitting of the otherwise degenerate two chiralities of its 120˚ magnetic structure. Our result is one of the most unusual cases of topological physics due to this broken $Z_2$ symmetry of the chirality in the supposedly paramagnetic state of YMnO$_3$, due to strong topological spin fluctuations with the additional intricacy of a Dzyaloshinskii-Moriya interaction.

Heat can, in principle, be carried by any quasiparticles, be they boson or fermion, but two such examples—phonons and electrons—are the most prominent. For magnetic insulators, magnons can be an alternative medium while electrons are out of the game. When phonons carry heat for magnetic insulators, it is well-known that the temperature dependence follows the Debye–Gruneissen formula. With magnons added to the picture, the variation of thermal transport with temperature can be changed quantitatively. However, it has been widely assumed in the community that it would still remain more or less the same at a qualitative level.

The most prominent feature of phonons and magnons as heat carriers is that they are charge-neutral. This feature significantly impacts the fundamental problem of heat conduction transverse to a thermal gradient by phonons and magnons, as there can be no Lorentz force upon applying a magnetic field. This starkly contrasts with electrical conduction and the resulting Hall effect of electron currents discovered by Edwin Hall in 1879[1]. This discovery of off-diagonal terms in electrical conductivity has proved of colossal importance. It has more recently revolutionized modern physics with phenomena such as the quantized integer and fractional Hall effects in certain high-quality two-dimensional materials[2].

Until recently, however, the chance of finding similar off-diagonal terms in thermal conductivity has seemed to be highly unlikely, if not impossible. There is no intuitive mechanism for a magnetic field to bend heat flow. However, nature has a surprise up her sleeve, with now a dozen reports that such an unexpected off-diagonal term in the thermal conductivity, the so-called thermal Hall effect (THE), is found in several classes of materials. A series of experiments followed the first

[1]Center for Quantum Materials & Department of Physics and Astronomy, Seoul National University, Seoul 08826, Republic of Korea. [2]Center for Correlated Electron Systems, Institute for Basic Science, Seoul 08826, Republic of Korea. [3]Department of Applied Physics, The University of Tokyo, Bunkyo-ku, Tokyo 113-8656, Japan. [4]Department of Physics, Tokyo Institute of Technology, Meguro-ku, Tokyo 152-8551, Japan. [5]Department of Physics, University of Warwick, Coventry CV4 7AL, UK. [6]Department of Physics, Ulsan National Institute of Science and Technology, Ulsan 44919, Republic of Korea. [7]RIKEN Center for Emergent Matter Science (CEMS), Wako, Saitama 351-0198, Japan. [8]Institute of Applied Physics, Seoul National University, Seoul 08826, Republic of Korea. [9]These authors contributed equally: Ha-Leem Kim, Takuma Saito, Heejun Yang. ✉e-mail: nagaosa@riken.jp; jgpark10@snu.ac.kr

report of THE in 2005 for $Tb_3Ga_5O_{12}$:[3] other notable examples include $Lu_2V_2O_7$, $Tb_2Ti_2O_7$, α-$RuCl_3$, $La_2CuO_4$-based high-temperature superconductors, and $SrTiO_3$, to name only a few[4–9]. Faced with the diverse class of materials showing a THE and their resulting different forms, it is not an unreasonable guess that there can be several different mechanisms (even competing mechanisms) at work for each of those materials with a visible sign of a THE[10]. Whatever mechanism is behind each experimental observation, it is clear now that THE reveals much of the hitherto neglected but fascinating nature of the charge-neutral heat current. For comparison with our work reported in this paper, it should be noted that most previous THE reports have been made for the ground and low-temperature ordered phases of given materials.

$YMnO_3$ is a type 1 multiferroic with a ferroelectric phase transition above 900 K and antiferromagnetic ordering at a much lower temperature of $T_N = 72$ K. The ferroelectric transition originates from a structural trimerization induced by the buckling of $MnO_5$ bipyramids[11], which is boosted via a strong spin–lattice coupling upon entering the antiferromagnetic phase[12]. The triangular lattice arrangement of magnetic $Mn^{3+}$ ($S = 2$) ions with antiferromagnetic Heisenberg interactions leads to strong spin fluctuations promoted by a significant geometrical frustration with the frustration ratio of 7, which is defined as the ratio between Curie–Weiss temperature ($\Theta_{CW}$) and $T_N$, i.e., $|\Theta_{CW}|/T_N$[13].

The presence of spin fluctuations is clear from powder neutron scattering data[13]. The suppression of thermal conductivity resulting from phonon scattering from the fluctuations has been reported[14]. The fluctuations are most considerable in magnitude at $T_N$ but still finite, even below and above the transition. Furthermore, $YMnO_3$ hosts a strong spin–lattice interaction of an exchange-striction type[15]. Theoretically, the magnon Hall effect could originate from the nontrivial band structure of magnons in the ordered phase[16]. However, it is essential to note that the two exchange integrals are not the same $J_1 \neq J_2$, where $J_1$ and $J_2$ are intra-trimer and inter-trimer exchange interactions due to the structural trimerization, respectively (Fig. 1a). What is essential for the discussion later is that spin-chirality does not cancel altogether for this trimerized triangular lattice with a noncollinear spin configuration, opening the door for a finite thermal Hall effect that is otherwise totally absent for a perfect triangular lattice.

In this work, we report the measurement of the thermal Hall effect in the supposedly paramagnetic phase of single-crystal $YMnO_3$, a well-studied hexagonal manganite insulator with frustrated antiferromagnetism. Thanks to our theoretical studies, we can attribute our measured THE signal to a new phenomenon induced by the nontrivial band topology of its spin excitations.

## Results and discussion

Two batches of high-purity $YMnO_3$ single crystals were cut into an $x$-$y$ planar aspect ($\mathbf{x} \parallel a$-axis), normal to the crystallographic $c$-axis. The magnetization data of the $YMnO_3$ single crystal and the Curie–Weiss fitting result are shown in Fig. 1b. A sharp peak at 72 K represents the antiferromagnetic transition. Below the 250 K, we can clearly see that the data starts to deviate from the Curie–Weiss behavior. The Curie–Weiss fitting produces $\Theta_{CW} = -497$ K and effective magnetic moment of 5.08 $\mu_B$, which is consistent with the previous study[17].

The thermal transport experimental setup is shown in Fig. 1c (see the "Methods" section for details). The result for longitudinal thermal conductivity ($\kappa_{xx}$) taken at zero fields is consistent with the previous report[14] (Fig. 2a). Measured longitudinal magneto-thermal conductivity (MTC), $\Delta\kappa_{xx}(H)$, for different sample temperatures, is shown in Fig. 2d. The MTC is <5% for the measurement range, indicating that the dominant heat carriers are phonons. The sign of the MTC is negative for temperatures below $T_N$. Interestingly, the reduction of the magnitude of $\kappa_{xx}$ is observed in our data with the applied field below $T_N$. The size of the MTC above $T_N$ was <0.1%, almost absent up to the resolution of our measurement system.

The temperature-dependent thermal Hall angle ratio (THAR), $\kappa_{xy}/\kappa_{xx}$, at 9 T is shown in Fig. 2a, where $\kappa_{xy}$ denotes thermal Hall conductivity. The THAR data for sample 2 were scaled by an empirical factor of 0.45 for plotting to give overlap with those from sample 1. This discrepancy mainly comes from the twofold difference of $\kappa_{xy}$, and we would like to note that the size of $\kappa_{xy}$ is known to be highly sensitive to the sample quality: even samples from the same source tend to exhibit variations in their magnitude. For example, for $Tb_2Ti_2O_7$, $SrTiO_3$, and Kitaev quantum spin liquid candidates, slightly different values of $\kappa_{xy}$ were reported by diverse researchers, but the sign, order of magnitude, and temperature or field dependences of $\kappa_{xy}$ were quite reproducible[5,6,8,9,18–27]. Likewise, THAR for samples 1 and 2 falls to a single curve after scaling, indicating the good reproducibility of our data. The observed THAR is several times smaller than the smallest values previously reported[22], demonstrating the extremely high precision of our setup. The temperature-dependent thermal Hall conductivity, $\kappa_{xy}$, at 9 T is plotted in Fig. 2b, with the same scaling factor for sample 2. At high temperatures, $\kappa_{xy}$ values for the two samples fall to a single curve. On the other hand, we see a slight deviation at temperatures below $T_N$. Note that $\kappa_{xy}$ is still finite above $T_N$. Finally, we note that the magnitude of $\kappa_{xy}$ increases linearly with the applied magnetic field (in analogy to electrical Hall effects) over the entire measured temperature range (Fig. 2c).

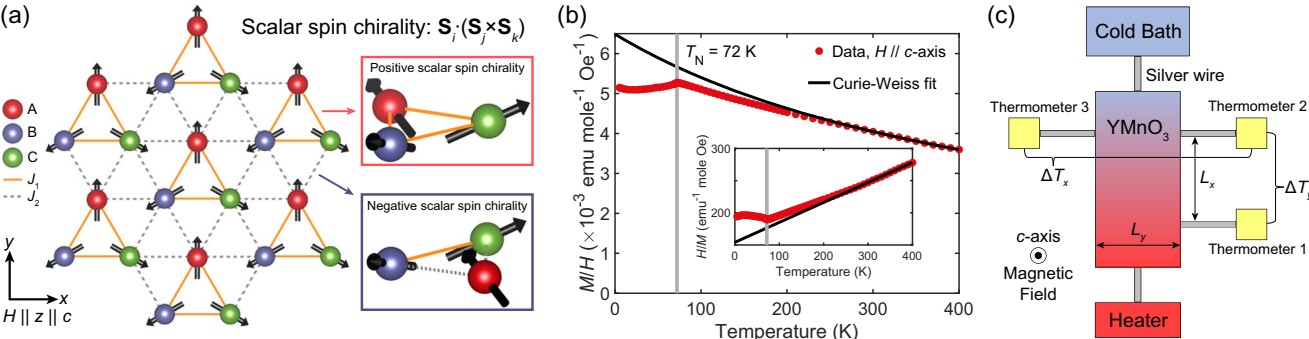

**Fig. 1 | Lattice structure, magnetization data of $YMnO_3$, and schematic of the thermal transport measurement setup. a** Schematic figure of the trimerized triangular lattice with a 120° magnetic structure. The three sites connected by thick orange lines consist of a unit cell. The magnetic field is applied along the $z$-direction. The red, blue, and green spheres represent sublattices $A$, $B$, and $C$. The exchange interactions $J_{ij}$ are $J_1$ and $J_2$ on the orange-solid and gray-dotted edges, respectively. The right side of the figure represents the enlarged picture of each triangle which can possess either positive or negative scalar spin chirality defined as $\mathbf{S}_i \cdot (\mathbf{S}_j \times \mathbf{S}_k)$ by the spin sites $i, j$, and $k$ in a counterclockwise way. **b** Measured magnetization ($M$) data of $YMnO_3$ single crystal sample as a function of the temperature. The magnetic field ($H = 2000$ Oe) was applied along the $c$-axis of the sample. A sharp peak at 72 K indicates an antiferromagnetic transition. The inset shows $H/M$ data with the black curves of the Curie–Weiss fitting results. **c** Schematic of the thermal Hall measurement setup.

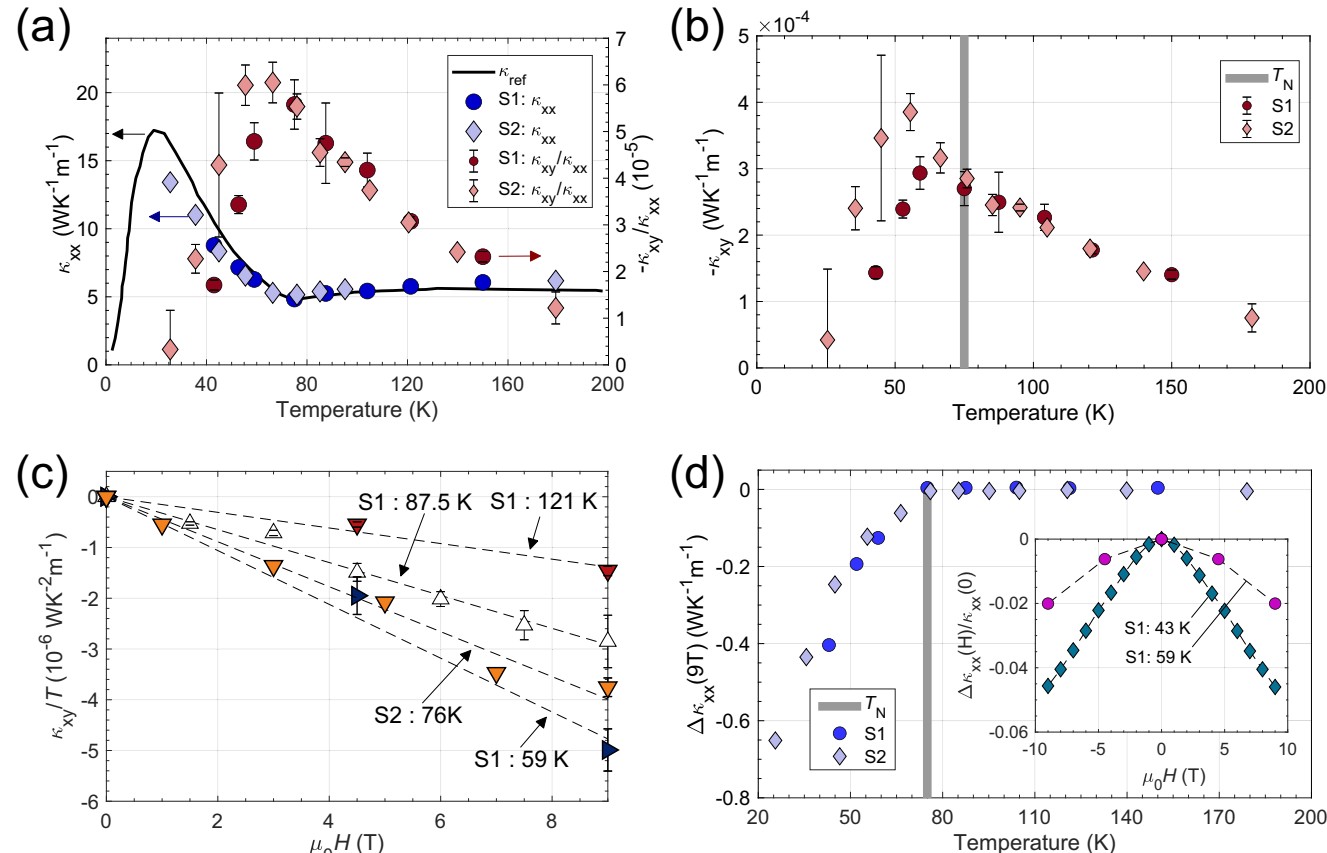

**Fig. 2 | Longitudinal thermal conductivity and thermal Hall conductivity.**
**a** Measured temperature-dependent thermal Hall angle ratio, $\kappa_{xy}/\kappa_{xx}$, at 9 T (red circle/diamond, right axis) and thermal conductivity $\kappa_{xx}$ taken at 0 T (blue circles/diamond, left axis). S1 and S2 indicate samples 1 and 2, respectively. THAR for sample 2 was scaled as 0.45 for reasons explained in the text. Zero field thermal conductivity data ($\kappa_{ref}$) from ref. 14. is also drawn as a **b**lack curve (left axis).
**b** Measured temperature-dependent thermal Hall conductivity, $\kappa_{xy}$, at 9 T. **c** $\kappa_{xy}/T$ for different sample temperatures plotted against the applied magnetic field. The data show good agreements to linear fits, shown as dashed lines (for **b** and **c**, $\kappa_{xy}$ for sample 2 was scaled as 0.45 likewise). **d** Temperature-dependent magnetothermal conductivity at 9 T, $\Delta\kappa_{xx}(9\text{T}) = \kappa_{xx}(9\text{T}) - \kappa_{xx}(0\text{T})$, with the thick line indicating the antiferromagnetic transition temperature, $T_N$. (inset) Normalized magnetothermal conductivity, $\Delta\kappa_{xx}(H) = [\kappa_{xx}(H) - \kappa_{xx}(0T)]/\kappa_{xx}(0T)$, for different temperatures. The broken curve/lines are a guide for the eyes. All error bars are standard deviations which were obtained by multiple measurements.

The above observations imply that this newly measured THE cannot be solely explained by the magnon Hall effect (MHE) as it is found to still exist in the supposedly paramagnetic phase. As noted previously, theoretical publications have predicted the existence of the MHE in this material in the ordered phase below $T_N$[16]. However, above $T_N$, the magnon is not well defined and thus $\kappa_{xy}$ should vanish according to the MHE scenario. Indeed, the MHE diminishes sharply above the magnetic ordering temperature, as demonstrated by thermal Hall measurements in $Lu_2V_2O_7$[4]. Furthermore, our longitudinal MTC data rule out a finite $\kappa_{xy}$ above $T_N$ induced by still-surviving weak magnons. Below $T_N$, we see a decrease of $\kappa_{xx}$, whilst right above $T_N$, the size of the MTC is hugely suppressed (~0.1%), indicating the sharp near-absence of any magnon contribution.

Given the material's strong spin–lattice coupling, the phonon Hall effect would be a natural guess for the mechanism responsible for THE above $T_N$. Notably, the thermal Hall angle ratio exhibits a maximal value at $T_N$, where spin fluctuations are most robust and decrease in magnitude at temperatures away from $T_N$ (Fig. 2a). From the observation, one might naively expect the observed THE to be due to phonons, assisted by the strong spin fluctuations.

For YMnO₃, the correlation length of spin fluctuations ($\xi_{spin}$) is 4–10 times larger than the wavelength of the phonons ($\lambda_{phonon}$) in the temperature range measured (at 100 K, $\lambda_{phonon}$ ~ 2–5 Å, $\xi_{spin}$ ~ 38 Å)[14]. It is reasonable to expect the scattering potential of spin fluctuations to be momentum-dependent too. Strong spatial correlations within the

spin fluctuations might also introduce further scattering between them and phonons. Presumably, the combined effect of convoluted momentum-dependent scattering and strong spin–lattice coupling could lead to a finite $\kappa_{xy}$. However, we stress that the above reasoning originating from the unusual behavior of the thermal Hall angle ratio (THAR) is misleading: $\kappa_{xx}$, which has a substantial local minimum at $T_N$, appears in the THAR's denominator. Thus, within the wide variation of the temperature dependence of $\kappa_{xy}$, the THAR will always have a local maximum at this temperature. Further detailed analysis shows that the skew-scattering scenario can be ruled out for the case of YMnO₃ (see Supplementary Note 2).

We analyzed theoretically the contribution to the thermal Hall conductivity from the spin system, which is described by the following Hamiltonian:

$$H = \sum_{\langle i,j \rangle} \left[ J_{ij}\mathbf{S}_i \cdot \mathbf{S}_j + \mathbf{D}_{ij} \cdot \left( \mathbf{S}_i \times \mathbf{S}_j \right) \right] - \sum_i \mathbf{h} \cdot \mathbf{S}_i + \sum_i \mathbf{S}_i^{\mathrm{T}} \Delta \mathbf{S}_i, \quad (1)$$

where $J_{ij}$ is the exchange interaction, $\mathbf{D}_{ij}$ is the Dzyaloshinskii–Moriya (DM) interaction. $\mathbf{h}$ is the external magnetic field, $\Delta$ is the single-ion anisotropy matrix which satisfies $\Delta^{\mathrm{T}} = \Delta$ (for more details, see the "Methods" section). We have calculated the temperature dependence of $\kappa_{xy}$ for this spin Hamiltonian in terms of the stochastic Landau–Lifshitz–Gilbert (LLG) equation combined with the Monte-Carlo simulation (see Supplementary Note 6 for details). The ground

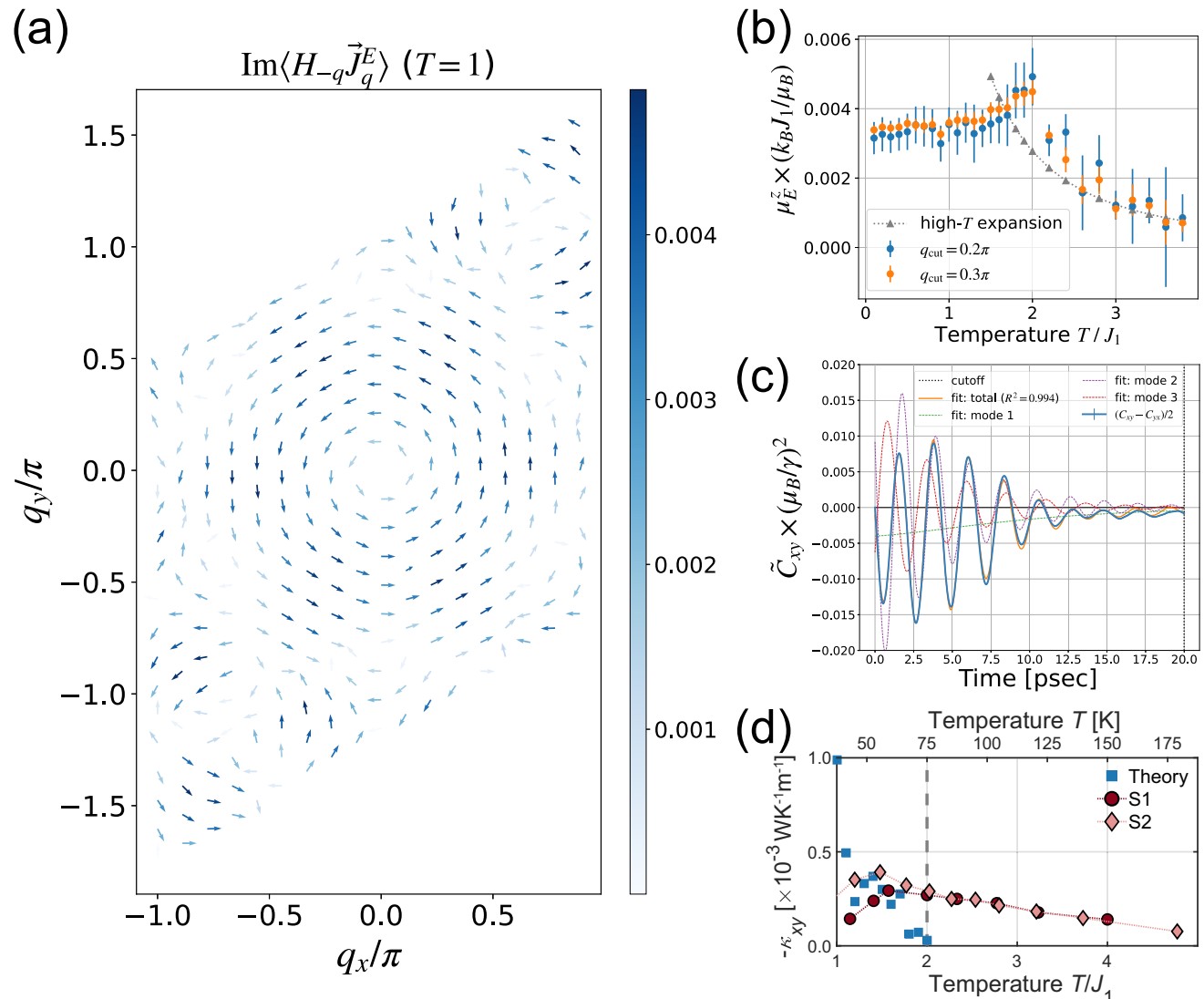

**Fig. 3 | The theoretical calculation of the thermal Hall conductivity.**
**a** $\mathrm{Im}\langle H_{-\mathbf{q}};\mathbf{J}_{\mathbf{q}}^{E}\rangle$ appearing in Eq. (4). One can see the vortex-like structure in the momentum space. The color bar is given in an arbitrary unit. **b** The temperature dependence of the z-component of $\boldsymbol{\mu}^{E}$ defined in Eq. (4). The triangular dotted curve shows the high-temperature expansion result, and the blue and red points show the numerically obtained results with different momentum cut-offs. **c** The

energy current correlation function $\widetilde{C}_{xy}(t)=(C_{xy}(t)-C_{yx}(t))/2$ with $C_{\mu\nu}(t)=\langle J^{E,\mu}(t)J^{E,\nu}(t)\rangle$ calculated by the stochastic LLG equation. The fitting by the superposition of the damped oscillators is also shown. **d** The obtained result of the thermal Hall conductivity as a function of temperature (solid blue squares) together with the experimental values for two samples. S1 and S2 indicate sample 1 and sample 2, respectively. All error bars are standard deviation from our calculations.

state is the 120° structure with a uniform z-component under the external magnetic field. The transition temperature is estimated from the peak of the specific heat as $T_{\mathrm{N}}\cong 2J_{1}$. As for the thermal Hall conductivity, there are two contributions as $\kappa_{xy}=\kappa_{xy}^{\mathrm{Kubo}}+\kappa_{xy}^{\mathrm{EM}}$, where

$$\kappa_{xy}^{\mathrm{Kubo}}=\frac{1}{k_{\mathrm{B}}T^{2}}\int_{0}^{\infty}dt\,\langle J^{E,x}(t);J^{E,y}(0)\rangle,\qquad(2)$$

and

$$\kappa_{xy}^{\mathrm{EM}}=\frac{2M^{E,z}}{VT},\qquad(3)$$

the latter of which can be obtained from[28]

$$-\partial_{\mathrm{T}}\left(\frac{\mathbf{M}^{\mathrm{E}}}{T^{2}}\right)=\frac{\boldsymbol{\mu}^{\mathrm{E}}}{T^{3}},\quad\boldsymbol{\mu}^{\mathrm{E}}:=\frac{1}{2iT}\nabla_{q}\times\left.\left\langle H_{-\mathbf{q}};\mathbf{J}_{\mathbf{q}}^{\mathrm{E}}\right\rangle_{\mathrm{eq}}\right|_{\mathbf{q}\to 0}.\qquad(4)$$

Here $J^{E,x}$ ($J^{E,y}$) is the energy current density along $x$ ($y$) direction, and the energy magnetization $\mathbf{M}^{E}$ is given by the Fourier component of the Hamiltonian $H_{-q}$ and that of $\mathbf{J}^{E}$. (The details of these formulas are given in Supplementary Note 5) As shown in Fig. 3a and b, we have estimated numerically $\boldsymbol{\mu}^{E},\mathbf{M}^{E}$ and $\kappa_{xy}^{\mathrm{EM}}$ by Monte Carlo simulation. The high-temperature expansion for $\langle H_{-\mathbf{q}};\mathbf{J}_{\mathbf{q}}^{E}\rangle_{\mathrm{eq}}$ is employed to obtain the asymptotic form and performed the integration with respect to $T$ from $T=\infty$ to $T=3J_{1}$ where the numerical data agrees with the asymptotic form, as shown in Fig. 3b. Figure 3d shows the obtained result for the numerically obtained $\kappa_{xy}$ (blue squares) together with the experimental data for two samples. At $T>2J_{1}$, $\langle J^{E,x}(t);J^{E,y}(0)\rangle$ becomes very small and buries in the thermal noise, indicating $\kappa_{xy}$ is invisible on this scale. Below $J_{1}$, the obtained $\kappa_{xy}$ depends strongly on the number of damped oscillators for the fitting, and we could not obtain the convergence.

The quantum effect should also be relevant at lower temperatures; hence, we did not present the numerical data in that temperature region. However, it is true that $\kappa_{xy}$ should go to 0 as the

temperature approaches 0, and $|\kappa_{xy}|$ should have a maximum below $T_N$. There are a few indications from comparing the theoretical results and the experimental data. One is that the order of magnitude of $\kappa_{xy}$ is comparable between the two. Since the parameters of the spin Hamiltonian are relatively well known in this material, the contribution from the spin system can be obtained without fitting parameters, and we conclude that the appreciable part of $\kappa_{xy}$ comes from the spins.

We have calculated the intrinsic thermal Hall effect due to phonons in the spin wave approximation to estimate the contribution from phonons. As a typical value of the spin–phonon Raman interaction ($\lambda$) in a 3$d$ transition metal atom system, we take the calculated value of phonon splitting in $CrI_3$ as $\lambda$ - 0.005 meV[29]. Taking this value of $\lambda$, the obtained result is shown in Supplementary Fig. 16, where the contribution is around -−3 × 10$^{-5}$ W K$^{-1}$ m$^{-1}$ at 100 K and around one order of magnitude smaller than that from spins in this low-temperature region. This is consistent with our conclusion that the spin contribution is dominant below $T_N$. However, higher energy phonons are thermally distributed as the temperature increases, and the extrinsic contribution is expected to grow. The calculated spin contribution decays more rapidly than in the experimental data above $T_N \cong 2J_1$. This indicates that the phonon contribution cannot be entirely neglected, especially at higher temperatures, i.e., the spin–phonon interaction and, e.g. the skew scattering of phonons by spins plays a crucial role. This is an exciting question and requires further detailed theoretical studies.

Another critical question is why it is so small compared with other materials[3]. One reason is that the uniform Raman interaction, which contributes to the thermal Hall effect, is proportional to the uniform magnetization $M_z$. $M_z$, which comes from the canting of spins from the AF state on the triangular lattice, is relatively small ($M_z \sim \frac{hS}{9}$ with $S = 2$ at $B = 9$ T). The other reason is that the spin–orbit interaction in the 3$d$ system is smaller than those in the 4$d$ and 5$f$ systems.

To conclude, we observed a finite thermal Hall effect induced by topological spin fluctuations in a trimerized triangular-lattice compound $YMnO_3$. Conventionally, the triangular lattice system is expected to have no thermal Hall effect due to chirality cancellation. However, in $YMnO_3$, via structural trimerization assisted by the DM interaction, chirality does not cancel altogether, resulting in a finite THE. This leads to an unexpected and exotic THE due to the topological properties of the spin fluctuations, captured by our theoretical calculation using the stochastic LLG equation. This topological nature of spin fluctuations we found is new and fundamental, making our result one of the most unusual cases of topological physics due to this broken $Z_2$ symmetry of the chirality in the supposedly paramagnetic state of $YMnO_3$. This new mechanism offers a new direction in exploring new thermal Hall physics and exotic excitations.

## Methods

### Sample preparation
$YMnO_3$ single crystal samples were synthesized by using a commercial optical floating zone furnace (Crystal Systems, Japan) following the recipe as described in the literature[15]. The magnetic susceptibility of the samples was measured by using a commercial setup (MPMS5XL, Quantum Design, USA).

### Thermal transport measurements
Therma transport measurements were conducted by steady-state method inside a commercial cryostat (PPMS9, Quantum Design USA) by using a home-built thermal Hall probe and measurement electronics system in a temperature range of 25–180 K, with the applied magnetic field up to 9 T ($\mathbf{H} \parallel \mathbf{z}$ and thermal current density $\mathbf{J}_q \parallel \mathbf{x}$). A temperature difference and hence fixed thermal gradient was set up along the $x$-direction, and temperatures were recorded to quantify $\Delta T_x$ and $\Delta T_y$; which allows extraction of the individual elements of the thermal conductivity tensor $\boldsymbol{\kappa}$ (Fig. 1c). $\kappa_{xy}$, the thermal conductivity transverse to the applied heat gradient, and $\kappa_{xx}$, the conventional longitudinal thermal conductivity along $x$, were measured simultaneously using three $SrTiO_3$ capacitance thermometers, whose main advantage is that we avoid magnetoresistance errors and do not need to perform extensive magnetic field calibrations. Our setup is also essentially free of thermometer self-heating effects. The detailed information on our thermal transport measurement setup is summarized in Supplementary Fig. 1[22].

The typical thermal Hall signal in $YMnO_3$ was <0.2 mK for all temperature ranges, making measurements challenging. Thus, special care has been taken to ensure the observed thermal Hall signal is intrinsic. For instance, a long time-scale drift of $\Delta T_y$ could be a major source of error; a spurious THE-like signal. The magnetic field was swept from positive to negative values before backing to the starting positive value for each measurement point to avoid such systematic errors. As $\kappa_{xy}$ must be an odd function of field direction analogously to electrical Hall signals, anti-symmetrized $\Delta T_y$ values were taken from an average of these positive and negative sweeps (Supplementary Note. 1). During $\kappa_{xy}$ measurements, the applied heater power was controlled so that $\Delta T_x$ was always kept smaller than 5% of the average sample temperature.

### Landau–Lifshitz–Gilbert calculations
For $YMnO_3$, $Mn^{3+}$ ions have the localized spin $S = 2$ forming a trimerized triangular lattice. The antiferromagnetic exchange interaction $J_{ij}$ takes two different values, i.e., intra-trimer ($J_1$) and inter-trimer ($J_2$) interactions, and we take the values $J_1 = 2.0$, $J_2 = 2.4$, and $\Delta^{zz} = 0.3$ meV according to ref. 16. As for the Zeeman coupling, $h = g\mu_B B$, with the $g$-factor $g = 2$ and the Bohr magneton $\mu_B = 5.788 \times 10^{-5}$ eV/T. The DM coupling $\mathbf{D}_{ij} = V_{DM}\mathbf{e}_z = 0.03J_1\mathbf{e}_z$ for the nearest-neighbor sites is chosen from the first-principle calculation[30]. From the viewpoint of symmetry, trimerization, Zeeman, and DM couplings are all necessary for finite thermal Hall conductivity $\kappa_{xy}$.

Our integration was done numerically for $T < 3J_1$. $\kappa_{xy}^{Kubo}$ is estimated from the time-dependent energy current correlation function $\langle J^{E,x}(t); J^{E,y}(0)\rangle$ obtained by stochastic LLG equation (Fig. 3c). We introduced the estimator of the errors and the cut-off time beyond which the data are discarded. We fitted the data by combining damped oscillators and performed the integral over time $t$. This fitting works well in the temperature region $J_1 < T < 2J_1$.

## Data availability
The datasets generated and/or analyzed during the current study are available from the corresponding author upon request. Source data for experimental results are provided with this paper. Source data are provided with this paper.

## Code availability
Custom codes used in this article are available from the corresponding author upon request.

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

## Acknowledgements

The work at SNU is funded by the Leading Researcher Program of the National Research Foundation of Korea (Grant No. 2020R1A3B2079375). T.S. was supported by JSPS KAKENHI Grant Number 21J13342, and N.N. was supported by JST CREST Grant Number JPMJCR1874, Japan. J.H.L. and Y.S.O. acknowledge support from the Basic Science Research Programs through the National Research Foundation of Korea (NRF) (No. RS-2023-00244404).

## Author contributions

J.-G.P. initiated and supervised the project. H.S. synthesized the single-crystal samples. H.Y. measured the magnetization of samples. H.-L.K. and H.Y. performed thermal transport measurements. H.-L.K., T.S., H.Y., H.I., M.J.C., J.H.L., Y.S.O., N.N., and J.-G.P. analyzed the data. T.S., H.I., and N.N. conducted thermal Hall conductivity calculations. T.S., H.I., N.N., and J.-G.P. contributed to the theoretical analysis and discussion. H.-L.K., T.S., H.Y., N.N., and J.-G.P. wrote the manuscript with contributions from all authors.

## Competing interests

The authors declare no competing interests.
