## [Peer Review File · Nature Communications]

REVIEWER COMMENTS

Reviewer #1 (Remarks to the Author):

This paper is certainly very interesting contribution to the rapidly developing and exciting field of the thermal Hall effect in correlated electron materials. It contains an impressive combination of experiment and theory, and is certainly deserving of publication in Nature Communications.

Some questions

(i) It is very impressive that the authors have also managed to compute the energy magnetization. However, it is not clear how this contributions cancels the $1/T$ singularity of the Kubo contribution, which is calculated from a different method.

(ii) More clarity is needed on the phonon contribution, as that appears to be quite important in other materials. The Hall angle here (10^{-5}) is smaller than in other phonon dominated materials (10^{-3}) so this may be an indication of the difference from other materials ?

Reviewer #2 (Remarks to the Author):

Review of Nature Comm. 428492

The Manuscript describes the measurements and MC calculations of primarily the Thermal Hall Effect in YMnO_3 single crystals. While I find the Manuscript interesting, I believe that there are some points that need to be addressed before it can be published. Below is a list of these for the attention of the Authors:

1. In Abstract: the phrase ‘...additional intricacy of DM interaction under an applied magnetic field.’ can be understood in at least a couple of different ways – this needs to be rephrased/explained.
2. The broken Z_2 symmetry, referred to in the Abstract, has to be illustrated visually – the type of topological fluctuations has to be explicitly referred to.
3. The spin-chirality that does not cancel altogether for the trimerized triangular lattice, should also produce deviations from the classical Curie-Weiss behavior of, for example, the paramagnetic

susceptibility that should be observable both computationally and experimentally. I would strongly encourage the Authors to include such data for their crystals.

4. There is something puzzling in the measurement geometry, as illustrated on say figure S1. The thermal contact for the measurement of the temperature of the hot end of the sample is brought onto one side of the sample only (left). This would create a small but likely non-negligible dynamic tilting of the isothermal lines through the crystal, which in turn can induce left-right antisymmetric thermal Hall-like signal.

5. The THAR data for the two samples has been scaled by an 'empirical factor' of ~ 0.5 . What is the justification for this procedure?

6. When the magnetic field is applied on the z-axis, there should be a finite projection of the spins on the same. What is the magnitude of this projection for the maximal fields applied (~ 9 T)? What does the magnetization data look like for this orientation of the applied field? Is there a spontaneous part of M_z ? Correspondingly, is there a spontaneous breaking of the Z2 symmetry?

7. If the THE effect can be associated with the small (potentially linear with applied field) M_z projection (or its sort-time average, at high temperature), then a clear and simple explanation of the proposed mechanism is needed.

8. Figure S3 is slightly misleading. As of the antisymmetrization (panel b), only half the points are independent, and the zero is automatically at zero. On panel a, the transitions from field are missing from the data set. These can be informative for the behaviour of the measurement system and have to be shown.

General comment: This paper is certainly very interesting contribution to the rapidly developing and exciting field of the thermal Hall effect in correlated electron materials. It contains an impressive combination of experiment and theory, and is certainly deserving of publication in Nature Communications.

Reply: We thank the referee for kindly recognizing the significance of our work.

Comment #1: It is very impressive that the authors have also managed to compute the energy magnetization. However, it is not clear how this contributions cancels the $1/T$ singularity of the Kubo contribution, which is calculated from a different method.

Reply: Thank you for asking this good question. Indeed, the cancellation of the $1/T$ divergence is not trivial. The argument is based on the spin wave theory, which predicts that the thermal Hall conductivity goes to 0 as T approaches 0 due to the Bose factor. In contrast, the classical spin wave theory predicts the finite value at $T=0$ (and hence the present classical simulation is expected to give the finite value).

In the present simulation results, the cancellation is not exact but is not contradicting it. As the temperature is lowered, both Kubo and energy magnetization contributions get larger, and hence the accuracy of the cancellation gets worse and worse. Therefore, we restricted our discussion relatively near the transition temperature, where the thermal Hall conductivity is appreciable.

Comment #2: More clarity is needed on the phonon contribution, as that appears to be quite important in other materials. The Hall angle here (10^{-5}) is smaller than in other phonon dominated materials (10^{-3}) so this may be an indication of the difference from other materials ?

Reply: We appreciate another excellent question from the referee. Estimating the phonon contribution is theoretically challenging since the coupling between spins and phonons is not known quantitatively and accurately enough. However, one can have a rough estimate of the so-called Raman interaction and the consequent contribution of the phonon to the intrinsic thermal Hall conductivity. Here we take the estimated value of the spin-phonon Raman interaction $\lambda \sim 0.005\text{meV}$ from the phonon splitting in CrI_3 as the typical value of 3d systems (John Bonini et al., Phys. Rev. Lett. 086701 (2023)) Taking this value of λ , the obtained result is shown in Fig.R1 below, where the contribution is around $\sim -3 \times 10^{-5}$ W/Km at 100 K, which is around one order of magnitude smaller than that from spins in this low-temperature region. This is consistent with our conclusion that the spin contribution dominates below T_N . However, as the temperature increases, higher energy phonons are thermally distributed and the extrinsic contribution is expected to grow.

It is also an important comment why it is so small compared with other materials. One reason is that the uniform Raman interaction, contributing to the thermal Hall effect, is proportional to M_z and $\text{SOI} \cdot M_z$, which comes from the canting of spins from the AF state on the triangular lattice, is relatively small ($M_z \sim hS / 9$). The other reason is that the SOI in the 3d system is smaller than those in the 4d and 4f systems. We have added this new calculation in the Supplementary Information and briefly mentioned it in the main text.

47
 48
 49
 50
 51
 52

Fig.R1: The phonon dispersion (left) and its thermal Hall conductivity (right) of the model calculation for YMnO₃. The intrinsic contribution to the thermal Hall conductivity is smaller than that from spins in this low-temperature region (see New Fig.S16).

53 -----
54 Report of the Second Referee -- NCOMMS-23-19843-T
55 -----

56
57 General comment: The Manuscript describes the measurements and MC calculations of primarily the
58 Thermal Hall Effect in YMnO₃ single crystals. While I find the Manuscript interesting, I believe that there
59 are some points that need to be addressed before it can be published. Below is a list of these for the
60 attention of the Authors:

61
62 **Reply:** We thank the referee for summarizing our work precisely. Following the valuable comments, we
63 revised the manuscript as below.

64
65 Comment #1: In Abstract: the phrase ‘...additional intricacy of DM interaction under an applied magnetic
66 field.’ can be understood in at least a couple of different ways – this needs to be rephrased/explained.

67
68 **Reply:** Following the referee’s advice, we rephrased it as “Here, we report that the trimerized
69 triangular lattice of multiferroic YMnO₃ produces a highly unusual thermal Hall effect due to
70 topological spin fluctuations in the presence of Dzyaloshinskii-Moriya interaction, under
71 applied magnetic field.”

72
73 Comment #2: The broken Z₂ symmetry, referred to in the Abstract, has to be illustrated visually – the
74 type of topological fluctuations has to be explicitly referred to.

75
76 **Reply:** Thank you for the excellent suggestion. The scalar spin chirality is induced by the
77 external magnetic field acting on the 120-degree spin structure in the trimerized triangular
78 lattice. The Z₂ symmetry is the two choices of the direction of the rotation of the spins, which
79 corresponds to the sign of the scalar spin chirality. Without trimerization, the scalar spin
80 chirality has the opposite sign in the neighboring triangles, while the trimerization violates this
81 cancellation. We have added this explanation in the abstract.

82
83 Comment #3: The spin-chirality that does not cancel altogether for the trimerized triangular lattice,
84 should also produce deviations from the classical Curie-Weiss behavior of, for example, the
85 paramagnetic susceptibility that should be observable both computationally and experimentally. I would
86 strongly encourage the Authors to include such data for their crystals.

87
88 **Reply:** Thank you for asking this nice question, which we overlooked previously. The figure below
89 represents the magnetic susceptibility data of our *h*-YMnO₃ single crystal sample, which shows a clear
90 deviation from the Curie-Weiss law around 250 K. We agree with the referee that the paramagnetic
91 susceptibility data is another supporting piece of information for our discussion of thermal Hall result.
92 We added our magnetic susceptibility data in the revised manuscript.
93

94
95
96
97
98
99
100
101

Fig. R2: A sharp peak at 72 K indicates the antiferromagnetic phase transition, and the Curie-Weiss fitting result is shown as the black curve in the inset of the figure. Below the 250 K, we can clearly see that our data deviate from the Curie-Weiss behavior. Our new results produce the Curie-Weiss temperature and effective magnetic moment of -497.4 K and $5.08 \mu_B$, respectively, which are consistent with the previous study [J. Park *et al.* Phys. Rev. B **82**, 054428 (2010)].

102
103
104
105
106
107
108
109

In response to the referee's comment, we have theoretically calculated the uniform magnetic susceptibility with and without the DM interaction, as shown in panel (a) of Fig.R3 below. Above T_N we find the Curie-Weiss (CW) behavior with negative CW temperature, while it deviates from CW behavior from slightly above T_N .

The effect of DM interaction is mainly the shift of T_N . Instead of the magnetic susceptibility, we can directly calculate the temperature dependence of the uniform scalar spin chirality in this trimerized triangular lattice, which is shown in panel (b) of Fig.R3. We have included these data in the SI.

110
111
112
113
114
115
116

Fig. R3: It shows the Temperature dependence of (a) the uniform magnetic susceptibility and (b) the scalar spin chirality of the Heisenberg model on a trimerized triangular lattice. The parameters of the model are the same as in the main text. The magnetic field is $B=9$ T both in panels (a) and (b)."

Comment #4: There is something puzzling in the measurement geometry, as illustrated on say figure S1. The thermal contact for the measurement of the temperature of the hot end of the sample is brought

117 onto one side of the sample only (left). This would create a small but likely non-negligible dynamic tilting
118 of the isothermal lines through the crystal, which in turn can induce left-right antisymmetric thermal Hall-
119 like signal.

120
121 **Reply:** Thank you for the comment. Indeed, there could be a left-right antisymmetric signal due to the
122 non-negligible dynamic tilting of the isothermal line. But the dynamic tilting effect will be symmetric with
123 respect to the magnetic field. Thus, as explained in the paper, the dynamic tilting-induced thermal Hall
124 signal will be cancelled out after the antisymmetrization of the raw data, a commonly used analysis
125 practice. We also want to point out that this five-contact geometry has been used in the community for
126 some time. For example, Ong's group at Princeton University used the same five-contact geometry to
127 measure the thermal Hall signal in a Kagome Magnet [see Fig. S2 in M. Hirschberger *et al.* Phys. Rev.
128 Lett. **115**, 106603 (2015)]. We have attached the measurement configuration of their sample below (Fig.
129 S2 on the paper).

130
131 As you can see, they also used only one thermal contact to measure the temperature of the hot end.
132 Ong's group has also used the same configuration in the other paper, for example, a recent
133 measurement of the thermal Hall effect on α -RuCl₃ [P. Czajka *et al.* Nat. Mater. **22**, 36–41 (2023)].

134
135 **Comment #5:** The THAR data for the two samples has been scaled by an 'empirical factor' of ~0.5.
136 What is the justification for this procedure?

137
138 **Reply:** As the referee pointed out, we scaled down the THAR of sample 2 by a factor of 0.45 for a better
139 comparison of both data: it is widely known that THAR is sample dependent even if one measures the
140 same materials. One can think of a few reasons. For instance, one source of errors could be ambiguity
141 for sample dimensions since the optical microscope could easily yield errors up to 10 % when we
142 estimate the length along x , y and z -directions.

143
144 Another equally possible reason is a sample quality issue: recent thermal Hall studies by several groups
145 found that the size of κ_{xy} is very sensitive to the sample quality. For example, as shown in the figure
146 below, the Kamran Behnia group reported a threefold difference of κ_{xy} in SrTiO₃ among different
147 samples, whilst the size of κ_{xx} is still similar for overall samples [for example, see Fig. 4 in X. Li *et al.*
148 Phys. Rev. Lett. **124**, 105901 (2020)]. We can note that the temperature dependence of κ_{xy} is robust
149 regardless of the samples, which implies that the signal is not an artefact.

150
151 Therefore we think that the twofold difference of κ_{xy} between our h -YMnO₃ samples might as well come
152 from slightly different sample quality. However, we would like to note that the robust temperature
153 dependence of κ_{xy} with the same sign is a crucial sign for the reproducibility of our experiment.

154
155 **Comment #6:** When the magnetic field is applied on the z -axis, there should be a finite projection of the
156 spins on the same. What is the magnitude of this projection for the maximal fields applied (~9 T)? What
157 does the magnetization data look like for this orientation of the applied field? Is there a spontaneous
158 part of M_z ? Correspondingly, is there a spontaneous breaking of the Z_2 symmetry?

159
160 **Reply:** There is no spontaneous symmetry breaking of M_z , which is induced by the external magnetic
161 field $h=H_z$. The applied magnetic field purely induces M_z , and there is no spontaneous symmetry
162 breaking of the Z_2 symmetry. Its magnitude is given by $M_z = S h / (3J_1 + 6J_2 + 3\sqrt{3}D+2\Delta)$. Using the
163 material parameters, the value of M_z at $B=9$ T is $M_z / S = 0.0489$.

164
165 **Comment #7:** If the THE effect can be associated with the small (potentially linear with applied field) M_z
166 projection (or its sort-time average, at high temperature), then a clear and simple explanation of the
167 proposed mechanism is needed.

168
169 **Reply:** Thank you for this sharp comment. The most convincing argument is based on the symmetry
170 consideration, which should apply to any microscopic mechanism. The symmetry related to THE of
171 spins is the product of time-reversal symmetry (TRS) and π -rotational symmetry of spins, which is often
172 called an effective TRS [PRB 99, 014427 (2019)].

173
174
175
176
177
178
179
180

Let's consider the case where no magnetic field is applied to our model. Then, all spins lie in the xy-plane, composing the 120-degree structure. Although the time-reversal operation flips the spins, the spin texture is equivalent to the original one because the two textures are connected by simultaneous π -rotation of spins. Once the magnetic field cants spin, the spin texture after the time-reversal operation can no longer be cancelled by the global rotation. As for the microscopic mechanisms, our simulation includes all of them, but the most probable one is the topological spin fluctuation associated with the scalar spin chirality.

181
182
183
184
185
186

Comment #8: Figure S3 is slightly misleading. As of the antisymmetrization (panel b), only half the points are independent, and the zero is automatically at zero. On panel a, the transitions from field are missing from the data set. These can be informative for the behaviour of the measurement system and have to be shown.

187
188
189

Reply: Thank you for the helpful comments. Based on the advice, we changed Figure S3(b) as below, showing only the positive field data. We also revised Figure 2(c) similarly to remove the redundancy of antisymmetrized data.

190
191
192
193
194

Fig. R4: New Fig. S3b.

Figure S3(a) shows the complete data, including data points between the field transition.

195
196

Fig. R5: New Fig. S3a

REVIEWERS' COMMENTS

Reviewer #1 (Remarks to the Author):

I am satisfied by the response of the authors